# Tillage System and Seeding Rate Impact on Yield, Oil Accumulation and Photosynthetic Potential of Different Cultivars of Safflower (*Carthamus tinctorius* L.) in Southern Russia

Sergey Voronov [1], Yurii Pleskachiov [1], Serazhutdin Kurbanov [2], Diana Magomedova [2,3] and Meisam Zargar [4,*]

[1] Department of Agronomy, Federal Research Center Nemchinovka, 121205 Moscow, Russia
[2] Department of Agrobiotechnology, Dagestan State Agrarian University, 367032 Makhachkala, Russia
[3] Department of Agronomy, Federal Agrarian Research Centre, Dagestan Republic, 367014 Makhachkala, Russia
[4] Department of Agrobiotechnology, Institute of Agriculture, RUDN University, 117198 Moscow, Russia
[*] Correspondence: author's zargar_m@pfur.ru

**Abstract:** Safflower (*Carthamus tinctorius* L.) is a high-value oilseed crop with growing importance in numerous countries around the globe. This study was performed to evaluate the efficacy of the basic tillage technique and various seeding rates on the yield, oil accumulation and photosynthetic potential of different safflower cultivars (Kamyshinskiy 73, Zavolzhskiy 1 and Alexandrite) in the Volgograd Region of Southern Russia. Three field experiments were conducted at the research farm of Volgograd Agricultural State University during 2018–2020. The treatments were arranged as split plots based on a randomized complete block design with three blocks. Tillage treatments (basic tillage as the control (moldboard PN-4-35, depth 20–22 cm), chisel ploughing (OCHO 5-40, depth 35–37 cm) and disc ploughing (BDM-4, depth 12–14 cm)) were assigned to the main plots, and seeding rates (200, 300 and 400 m$^2$/m$^2$) were allocated to the subplots. The maximum leaf area, with a value of 26.35 m$^2$/m$^2$ and the greatest photosynthetic potential of 1489 thousand m$^2$ × day ha$^{-1}$, was obtained in Alexandrite with the interaction of deep chisel ploughing and a seeding rate of 400 thousand seeds ha$^{-1}$. The highest dry biomass was also achieved in Alexandrite, with a value of 3.24 t ha$^{-1}$, with the interaction of deep chisel ploughing and a seeding rate of 400 thousand seeds ha$^{-1}$. The highest yield (1.84 t ha$^{-1}$) and oil accumulation (28.75%) were recorded for Alexandrite with the interaction of deep chisel ploughing and a seeding rate of 300 and 400 thousand seeds ha$^{-1}$ respectively. Overall, in terms of tillage treatments, the safflower productivity was highest when chisel tillage was performed, and the lowest yield was observed with the small disc ploughing practice. The yield of Alexandrite cultivar was noted to be 4.4–4.8% higher than that of the Zavolzhsky cultivar and 9.2–10.8% higher than that of the Kamyshinsky 73 cultivar.

**Keywords:** safflower yield; tillage system; seeding rate; leaf area; photosynthetic potential

## 1. Introduction

Safflower (*Carthamus tinctorius* L.) is an important crop that is well adapted to semiarid areas, and it is a day-neutral, long-day plant [1,2] that is grown in different seasons around the globe. Safflower is also adapted to the longer growing season and warmer conditions found in the central and southern parts of the Russian Federation [3]. Currently, the total area dedicated to safflower production all over the world is approximately more than 816,000 ha$^{-1}$, which constitutes less than 0.3% of the total area sowed to oilseed crops worldwide [1]. Recently, safflower has become a significant oilseed crop with a favorable oil content and fatty acid composition in arid and semiarid regions of the globe [4,5].

To date, the cultivation of oilseeds and the production of oil are among the highest priorities of the agricultural production and processing industry in Russia [6,7]. As of

late, agricultural producers have a noticeably increased interest in safflower because of its increased demand and lucrative prices. The conjecture for safflower is very high, and it has now become an opportunistic commodity. Processed products from safflower have begun to play a prominent role and occupy a high place in the Russian oil complex [8,9].

Many factors influence the productivity of the safflower plant, but the most important are fertilization, soil moisture [10], the seeding rate [11] and the tillage system [12]. Enhancing the seeding rate is one of the most significant ways to exploit the crop yield potential in farming systems. One of the reasons for the resultant crop yield improvement is the optimization of the plant density [13,14]. The plant density may affect both the vegetative and generative development in cultivated plants. The plant density significantly varies with environmental conditions and cultivation practices [14]. According to its plant traits, safflower can compensate for spatial variation by producing secondary and tertiary heads with more branches, so the plant morphology depends on the degree of intraspecific competition [15,16].

The aim of tillage in farming systems is to provide proper physical conditions for seed germination and plant growth [17,18]. An intensive tillage operation may lead to the disintegration of the soil structure because of the gradual loss of stable aggregates, leading to soil erosion and compaction that will cause low moisture availability for crops [19,20]. Tillage practices provide the most excellent opportunity to decrease the degradation of soil reserves and to increase soil productivity [21,22]. In order to ameliorate the water retention capacity of soils in semiarid regions, different tillage methods play a vital role in avoiding soil degradation by compaction [18,23]. In some soil conditions, different ploughing methods produce a wide range of results with respect to the hydraulic conductivity and bulk density of the soil in agricultural lands [24,25].

However, there is no available information on the interactive efficacy of tillage systems and seeding rates on safflower cultivation. The basic tillage system and the seeding rate or density were evaluated to have a great efficacy on safflower oil accumulation and yield. Experiments were carried out to study the effect of different tillage practices and seeding rates on the yield, oil accumulation and photosynthetic potential of different safflower cultivars in the conditions of the Volgograd Region of Southern Russia.

## 2. Materials and Methods

### 2.1. Site Description and Management

This study was conducted to evaluate the effect of various tillage practices and different seeding rates on yield, oil accumulation and photosynthetic potential of the various safflower cultivars. Three field trials were conducted at the research farm of Volgograd Agricultural State University, Volgograd, Russia (39.07° N, 44.17° E, and altitude of 1255 m above sea level) in 2018–2020 agricultural seasons. The climate was characterized by mean annual precipitation of 355.7 mm, mean annual temperature of 15 °C, mean annual maximum temperature of 17.6 °C and mean annual minimum temperature of 5.2 °C. The experimental site was located in a semiarid climatic zone in Southern Russia.

Three weeks before experiments, soil samples were randomly taken from all replications of each experiment to evaluate the physical and chemical properties. Hence, combined soil samples were collected at a depth of 0–35 cm in three experimental areas. The samples were dried at 60 °C, ground up, and analyzed according to standard procedure (Clemson University Agricultural Service Laboratory, Clemson, SC, USA). The soil was classified as loamy (loamy, thermic Typic Kandiudults) with a pH of 7.4 and an organic matter of 2.3%.

Appropriate fertilizers were applied, guided by the results obtained from the soil test to ensure proper crop growth of safflower. Basal fertilizer $N_{12}$–$P_{12}$–$K_{36}$ was applied at a rate of 260 kg ha$^{-1}$ to the experimental fields. At the safflower growth stage, 140 kg N ha$^{-1}$ (as urea) was applied as top dressing.

*2.2. Experimental Design*

The treatments were arranged as split plots based on a randomized complete block design with three replications. Basic tillage treatments (basic tillage as control (moldboard model PN-4-35, depth 20–22 cm), chisel ploughing (model OCHO 5-40, depth 35–37 cm) and disc ploughing (model BDM-4, depth 12–14 cm)) were assigned to the main plots and seeding rates (200, 300 and 400 thousand/ha$^{-1}$) were allocated to the subplots.

*2.3. Crop Management*

Standard production practices were adhered to in growing the crops. The soil was tilled with a field cultivator prior to planting. Following cultivation, 260 kg ha$^{-1}$ of fertilizer with an N–P–K ratio of 12–12–36 was side-banded or mid-row-banded in the seed rows for all plots. At safflower growth stage, 120 kg N ha$^{-1}$ (urea) top dressing was applied.

After land preparation, safflower seeds in different seeding rates were planted by a "Gaspardo" seeder on 22 April 2018, 15 April 2019 and 19 April 2020 at three cm depth. The area of each plot was 240 m$^2$ with a length of 20 m and a 12 m width. The area of each subplot was 72 m$^2$ with a length of 20 m, a 3.6 m width and a row spacing of 25 cm. There were two paths that were 0.6 m wide between subplots. The area under plots of each cultivar was 1944 m$^2$, and the total area under plots of all three cultivars was 5832 m$^2$. Safflower plants in all experimental plots were irrigated based on the local evapotranspiration rate of 5.5 mm day$^{-1}$ through a drip irrigation system distributed along the crop rows. Hand weeding was performed in all experimental plots during the growing season.

*2.4. Data Recording*

Each year, leaf area index (LAI), photosynthetic potential, safflower yield and seed oil percentage were evaluated and analyzed. Yield was determined by the harvest of three central rows in the last week of August in all three years of the experiments. Soxhlet extractor was used to evaluate the total oil accumulation of the safflower seeds, where 15 g of the milled seeds of safflower were put in extraction paper, and the oils were extracted in three hours using 300 mL of petroleum benzene. Oils were filtered with two filter papers, Buchner funnel and suction flask. The oil accumulation was recorded using the percentage proposed by Mohammadi et al. [1].

Leaf area index of the safflower plants was calculated by the use of a leaf area meter (CI-202 Area Meter, CID, Inc., Camas, WA, USA), and then plants were oven dried for 72 h at 75°C as demonstrated by Kaleem et al. [26]. The photosynthetic potential of the plants was measured using a photosynthesis meter (Cl-340, CID, USA). For this purpose, the fully expanded young leaves of the plants (from the middle row of each plot) were selected.

Safflower yield and yield parameters were recorded in all plots of each experiment and were taken randomly from two quadrats of 0.25 m$^2$ in each plot.

*2.5. Statistical Analysis*

Data were statistically analyzed using GLIMMIX procedure of SAS (Version 9.4, SAS Institute Inc., Cary, NC, USA), a mixed procedure where replications were considered as the random factor. The least squares mean statement in SAS with the Tukey adjustment at $p = 0.05$ was performed for the mean comparisons. Different tillage operations and various safflower seeding rates were investigated as fixed impacts when present in all experiments. The effect of experimental factors on yield, oil accumulation, leaf area and photosynthetic potential were statistically evaluated.

**3. Results**

According to the results, the interactions between the effects of the year and the treatment were not significant for the observed traits. The results of this study indicated that the leaf area of the Zavolzhsky 73 cultivar ranged from 17.10 to 25.14 m$^2$/m$^2$ for the interaction of disc ploughing × a seeding rate of 200 thousand seeds ha$^{-1}$ and deep

chisel ploughing $\times$ a seeding rate of 400 thousand seeds ha$^{-1}$, respectively, which was 0.73–1.36 thousand m$^2$ ha$^{-1}$ higher than that of the Kamyshinsky 73 cultivar. The maximum leaf area of the Alexandrite cultivar ranged from 17.94 m$^2$/m$^2$ for the interaction of disc ploughing $\times$ a seeding rate of 200 thousand seeds ha$^{-1}$ to 26.35 thousand m$^2$ ha$^{-1}$ for the interaction of deep chisel ploughing $\times$ a seeding rate of 400 thousand seeds ha$^{-1}$, which turned out to be 0.84–1.21 thousand m$^2$ ha$^{-1}$ more than that of the Zavolzhsky cultivar and 1.57–2.57 thousand m$^2$ ha$^{-1}$ higher than that of the Kamyshinsky 73 cultivar (Table 1).

**Table 1.** Interactive efficacy of the various tillage practices and seeding rates on safflower maximum leaf area index (LAI) average in 2018–2020.

| Tillage Practices | Seeding Rate (Thousand ha$^{-1}$) | Cultivars | | |
|---|---|---|---|---|
| | | Kamyshinskiy 73 | Zavolzhskiy 1 | Alexandrite |
| | | m$^2$/m$^2$ | | |
| Basic tillage ('control', moldboard PN-4-35, depth 20–22 cm) | 200 | 19.72 cd | 20.46 c | 21.97 d |
| | 300 | 21.39 bc | 22.43 b | 23.81 bc |
| | 400 | 20.61 c | 21.54 bc | 23.02 c |
| Chisel ploughing (OCHO 5-40, depth 35–37 cm) | 200 | 21.96 bc | 22.89 b | 24.17 b |
| | 300 | 23.78 a | 25.14 a | 26.35 a |
| | 400 | 22.73 b | 24.07 ab | 25.28 ab |
| Disc ploughing (BDM-4, depth 12–14 cm) | 200 | 16.37 ef | 17.10 de | 17.94 f |
| | 300 | 17.76 e | 18.49 d | 19.47 e |
| | 400 | 16.92 ef | 18.03 d | 18.44 ef |
| *p*-value | - | 0.02 | 0.009 | 0.001 |
| Coefficient of Variation (%) | - | 9.09 | 5.19 | 7.66 |

Means followed by different letters are significantly different by Tukey adjusted means comparisons at $p \leq 0.05$.

The photosynthetic potential of the Kamyshinsky 73 variety was attained in the range of 925 thousand m$^2$ $\times$ day ha$^{-1}$ for the interaction of the disc ploughing practice $\times$ a seeding rate of 200 thousand seeds ha$^{-1}$ to 1343 thousand m$^2$ $\times$ day ha$^{-1}$ for the interaction of the deep chisel ploughing treatment $\times$ a seeding rate of 400 thousand seeds ha$^{-1}$. The photosynthetic potential of the Zavolzhsky cultivar was achieved in the range of 966 and 1420 thousand m$^2$ $\times$ day ha$^{-1}$ when the disc ploughing treatment with a seeding rate of 200 thousand seeds ha$^{-1}$ and the deep chisel treatment with the seeding rate of 400 thousand seeds ha$^{-1}$ were performed, respectively, which was 41–77 thousand m$^2$ $\times$ day ha$^{-1}$ more than that of the Kamyshinsky 73 variety during the experimental years. The photosynthetic potential of the Alexandrite variety ranged from 1013 to 1489 thousand m$^2$ $\times$ day ha$^{-1}$ for the treatments of disc ploughing with a seeding rate of 200 thousand seeds ha$^{-1}$ and deep chisel ploughing with a sowing rate of 400 thousand seeds ha$^{-1}$, respectively, i.e., it was 47–69 thousand m$^2$ $\times$ day ha$^{-1}$ higher than the Zavolzhsky cultivar and 88–146 thousand m$^2$ $\times$ day ha$^{-1}$ more than the Kamyshinsky 73 cultivar (Table 2).

The dry biomass of the Kamyshinsky 73 cultivar that was obtained was in the range of 1.89 t ha$^{-1}$ for the interaction of the disc ploughing treatment with a seeding rate of 200 thousand seeds ha$^{-1}$ to 2.52 t ha$^{-1}$ for the interaction of the chisel ploughing option with a seeding rate of 400 thousand seeds ha$^{-1}$. A dry biomass ranging from 2.17 to 2.91 t ha$^{-1}$ was obtained for the Zavolzhsky cultivar using the disc ploughing treatment with a seeding rate of 200 thousand seeds ha$^{-1}$ and using the chisel processing treatment with a seeding rate of 400 thousand seeds ha$^{-1}$, respectively; it was 0.28–0.39 t ha$^{-1}$ higher than that of the Kamyshinsky 73 cultivar. In the Alexandrite cultivar, a dry biomass ranging from 2.41 t ha$^{-1}$ for the disc ploughing option with a sowing rate of 200 thousand seeds ha$^{-1}$ to 3.24 t ha$^{-1}$ for the chisel ploughing treatment with a sowing rate of 400 thousand seeds ha$^{-1}$ was obtained, which was a 0.24–0.35 and 0.52–0.72 t ha$^{-1}$ higher dry biomass than those in the Zavolzhsky and Kamyshinsky 73 cultivars, respectively (Figure 1).

**Table 2.** Interactive efficacy of the various tillage practices and seeding rates on photosynthetic potential of safflower average in 2018–2020.

| Tillage Practices | Seeding Rate | Varieties | | |
|---|---|---|---|---|
| | | Kamyshinskiy 73 | Zavolzhskiy 1 | Alexandrite |
| | | Thousand m$^2$ × Day ha$^{-1}$ | | |
| Basic tillage ('control', moldboard PN-4-35, depth 20–22 cm) | 200 | 1114 de | 1156 d | 1241 c |
| | 300 | 1208 bc | 1267 c | 1345 b |
| | 400 | 1164 d | 1217 cd | 1300 b |
| Chisel ploughing (OCHO 5-40, depth 35–37 cm) | 200 | 1241 b | 1293 bc | 1489 a |
| | 300 | 1343 a | 1420 a | 1489 a |
| | 400 | 1284 b | 1359 b | 1438 a |
| Disc ploughing (BDM-4, depth 12–14 cm) | 200 | 925 f | 966 f | 1013 de |
| | 300 | 994 e | 1045 e | 1100 d |
| | 400 | 956 ef | 1018 e | 1042 d |
| *p*-value | – | 0.005 | 0.088 | 0.001 |
| Coefficient of Variation (%) | – | 10.10 | 8.09 | 6.77 |

Means followed by different letters are significantly different by Tukey adjusted means comparisons at $p \leq 0.05$.

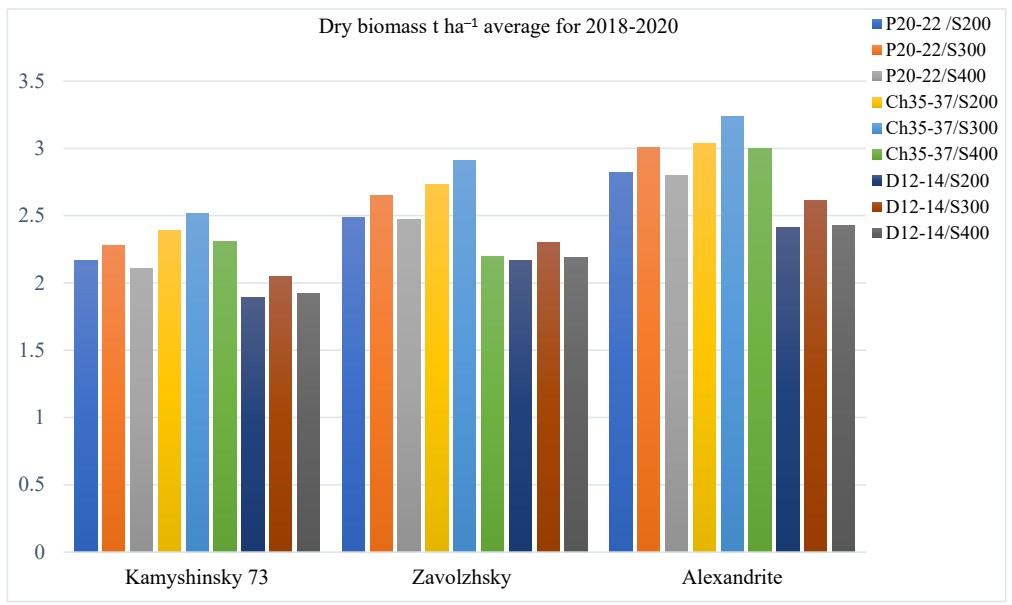

**Figure 1.** Interactive effect of different tillage techniques and seeding rates on dry biomass of safflower. Abbreviations: P20–22/S200 represents moldboard plow PN-4-35 with depth of 20–22 cm/seeding rate of 200; P20–22/S300 represents moldboard plow PN-4-35 with depth of 20–22 cm/seeding rate of 300; P20–22/S400 represents moldboard plow PN-4-35 with depth of 20–22 cm/seeding rate of 400; Ch35–37 cm/S200 represents chisel plow OCHO 5-40 with depth of 35–37 cm/seeding rate of 200; Ch35–37 cm/S300 represents chisel plow OCHO 5-40 with depth of 35–37 cm/seeding of rate 300; Ch35–37 cm/S400 represents chisel plow OCHO 5-40 with depth of 35–37 cm/seeding rate of 400; D12–14/S200 represents disc plow BDM-4 with depth of 12–14 cm/seeding rate of 200; D12–14/S300 represents disc plow BDM-4 with depth of 12–14 cm/seeding rate of 300; and D12–14/S400 represents disc plow BDM-4 with depth of 12–14 cm/seeding rate of 400.

An analysis of the safflower yield of the Kamyshinsky 73 cultivar indicated that, on average, in the trial period 2018–2020, according to the tillage treatment, the highest safflower yield was observed in the interaction of chisel ploughing at a depth of 35–37 cm, whereas the lowest yield was attained with the disc ploughing practice. According to the seeding rate, the highest and the lowest yields were obtained when the safflower was sown at seeding rates of 300 and 200 thousand seeds ha$^{-1}$, respectively. The maximum yield of 1.57 t ha$^{-1}$ was obtained with deep chisel ploughing with a seeding rate of 300 thousand

seeds per hectare, and the lowest yield of 1.08 t ha$^{-1}$ was obtained with the treatment of small disc ploughing with a seeding rate of 200 thousand seeds per hectare (Figure 2).

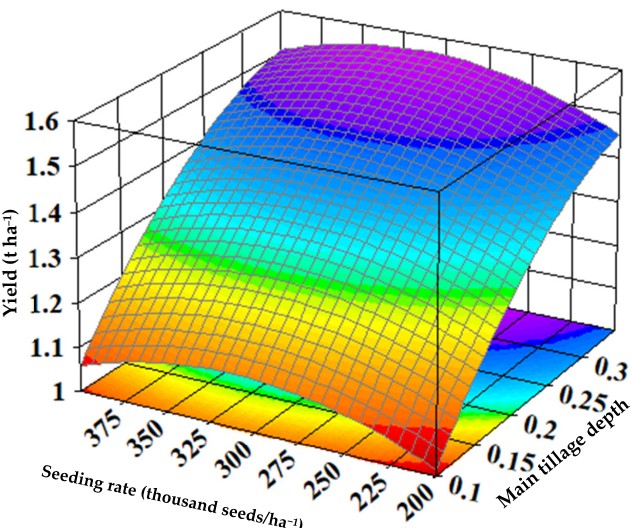

**Figure 2.** Response surface graph for the dependence of the yield of Kamyshinsky 73 safflower cultivar on seeding rate and tillage treatment.

For the seeding rate treatment, the Zavolzhsky cultivar yield was the highest when a seeding rate of 300 thousand seeds ha$^{-1}$ was adopted, and the lowest yield was obtained when 200 thousand seeds ha$^{-1}$ were planted. The maximum yield of 1.66 t ha$^{-1}$ was achieved for the Zavolzhsky safflower cultivar in the interaction of deep chisel ploughing $\times$ a seeding rate of 300 thousand seeds ha$^{-1}$. The lowest yield of 1.13 t ha$^{-1}$ was recorded for small disc ploughing with a sowing rate of 200 thousand seeds ha$^{-1}$ (Figure 3).

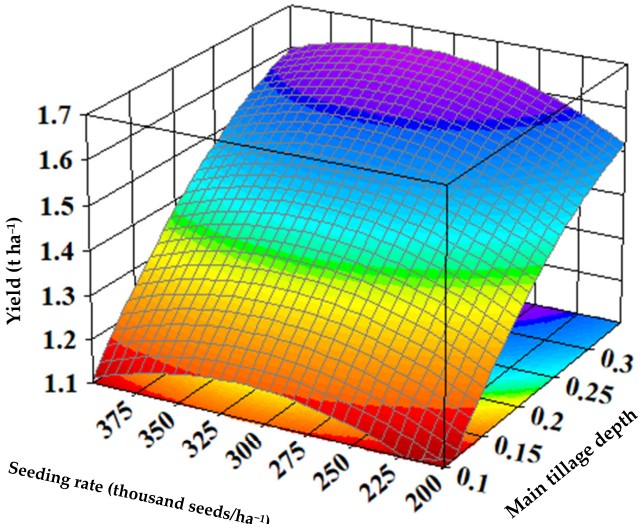

**Figure 3.** Response surface graph for the dependence of the yield of Zavolzhsky safflower cultivar on seeding rate and tillage treatment.

Our results showed that the yield of the Alexandrite cultivar was 4.4–4.8% higher than that of the Zavolzhsky cultivar and 9.2–10.8% higher than that of the Kamyshinsky 73 cultivar. Referring to the tillage practices, the yield was the highest for chisel ploughing at a depth of 35–37 cm, and the lowest yield was obtained when small disc tillage was executed. The results of the seeding rate indicated that the yield increased for sowing rates of 300 thousand seeds ha$^{-1}$, and the lowest yield was observed for sowing rates of 200 thousand seeds ha$^{-1}$ (Figure 4). The interaction of deep chisel ploughing with a sowing rate of 300 thousand seeds ha$^{-1}$ significantly increased the yield of the Alexandrite safflower cultivar to 1.74 t ha$^{-1}$, and, on the other hand, the minimum yield of 1.18 t ha$^{-1}$ was recorded for the disc plow treatment with a sowing rate of 200 thousand seeds ha$^{-1}$ (Figure 4).

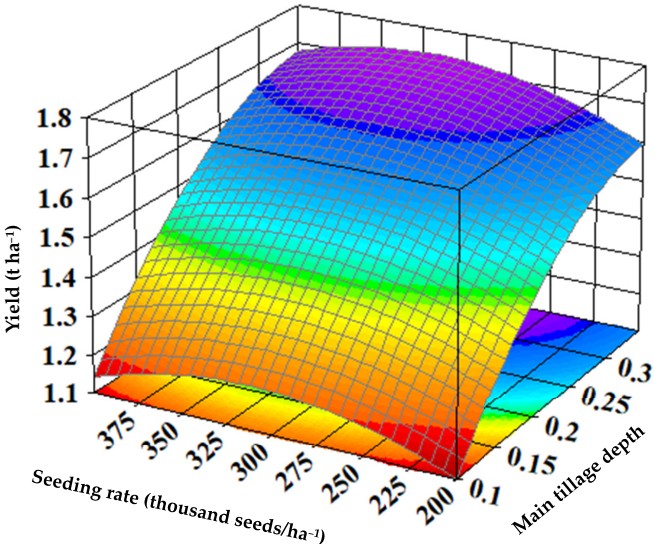

**Figure 4.** Response surface graph for the dependence of the yield of Alexandrite safflower cultivar on seeding rate and tillage treatment.

In addition to the yield of the Kamyshinsky 73, Zavolzhsky and Alexandrite cultivars, the correlation relationships, or dependencies, of the yield of safflower with the seeding rate and the depth of the main tillage were illustrated. The general form of the dependence of the safflower yield on various factors, such as the appropriate tillage depth and sowing rate, fully corresponds to the following polynomial of the second degree:

$$Y = a + b{\cdot}h + c{\cdot}S_r + d{\cdot}h^2 + e{\cdot}S_r^{\,2} + f{\cdot}h{\cdot}S_r$$

where $Y$ is the safflower yield, t ha$^{-1}$; $h$ is the depth of the ploughing of the soil corresponding to the factors of the processing technology in the experiments; $S_r$ is the sowing rate of safflower, thousand seeds ha$^{-1}$; and $a, b, c, d, e$ and $f$ are the parameters of the regression equation that determines the type of response surface.

The evaluation of the dependence parameters with the regression analysis method made it possible to establish the response surface equations for each of the safflower cultivars in the experiments. For the Kamyshinsky 73 safflower cultivar, the response surface equation is:

$$Y = -0.034 + 4.02{\cdot}h + 0.005{\cdot}S_r - 5.13{\cdot}h^2 - 8.1{\cdot}10^{-6}{\cdot}S_r^{\,2} + 1.9{\cdot}10^{-4}{\cdot}h{\cdot}S_r$$

The coefficient of determination of the dependence was 0.95, and the evaluated indicators were closely related.

Figure 2 shows the type of response surface for the dependence of the yield of the Kamyshinsky 73 cultivar. As exhibited in the figure, there was a significant trend in yield enhancement with an increase in tillage depth. A significant increase in the yield was noted even when developing a layer of soil that obviously exceeded the depth of the arable horizon. This, in general, is not a characteristic trend, and the resulting advantage denoted the effectiveness of the proposed method of soil cultivation that combined deep tillage with the turnover of half of the arable layer.

For the Zavolzhsky cultivar, the response surface equation is:

$$Y = -0.042 + 3.82 \cdot h + 0.005 \cdot S_r - 4.75 \cdot h^2 - 9.0 \cdot 10^{-6} \cdot S_r^2 + 6.3 \cdot 10^{-4} \cdot h \cdot S_r$$

The coefficient of dependence determination was 0.90, thus indicating a good convergence of the distribution of experimental data and the response surface. Figure 3 shows that the optimal level of the sowing rate corresponded to 300 thousand seeds ha$^{-1}$. This pattern was particular for the studied cultivars and did not depend on the accepted method of basic tillage.

For the Alexandrite safflower cultivar, the response surface equation is:

$$Y = -0.111 + 5.10 \cdot h + 0.006 \cdot S_r - 7.33 \cdot h^2 - 9.2 \cdot 10^{-6} \cdot S_r^2 + 8.4 \cdot 10^{-4} \cdot h \cdot S_r$$

The coefficient of determination was 0.93, the resulting equation described the distribution of experimental data under the established limits well. A graph of the response surface is shown in Figure 4. The general patterns established for the previously evaluated cultivars in Alexandrite were also preserved. The area of formation of the most productive crops corresponded to the combination of a sowing rate of 300 thousand seeds ha$^{-1}$ and deep basic tillage. The general position of the response surface for the Alexandrite cultivar was slightly higher than for the other cultivars studied in the experiments. This indicated that the greatest responsiveness to the ongoing agrotechnical activities, as well as the advantage in the potential of productivity in the typical conditions of the region, was in the Alexandrite cultivar.

In Table 3, the parameters of the response equation for the dependence of the safflower yield on the seeding rate and the tillage system were systematized relative to the cultivars evaluated in our study. Based on the results obtained, the safflower oil content depended on the various cultivars, seeding rates and different basic tillage practices, and it also changed over the years. On average, for 2018–2020, the oil percentage ranged from 26.12% in the Kamyshinsky 73 cultivar produced with disc ploughing with a sowing rate of 400 thousand seeds ha$^{-1}$ to 28.75% in the Alexandrite cultivar produced under the chisel ploughing practice with a sowing rate of 400 thousand seeds ha$^{-1}$ (Figure 5).

**Table 3.** Parameters of the response equation for the dependence of the yield of safflower on the seeding rate and tillage practices.

| Safflower Cultivars | Parameters of the Equation | | | | | | R$^2$ |
| --- | --- | --- | --- | --- | --- | --- | --- |
| | *a* | *b* | *c* | *d* | *e* | *f* | |
| Kamyshinskiy 73 | −0.034 | 4.02 | 0.005 | 5.13 | 8.1·10$^{-6}$ | 1.9·10$^{-4}$ | 0.95 |
| Zavolzhskiy 1 | −0.042 | 3.82 | 0.005 | 4.75 | 9.0·10$^{-6}$ | 6.3·10$^{-4}$ | 0.90 |
| Alexandrite | −0.111 | 5.10 | 0.006 | 7.33 | 9.2·10$^{-6}$ | 8.4·10$^{-4}$ | 0.93 |

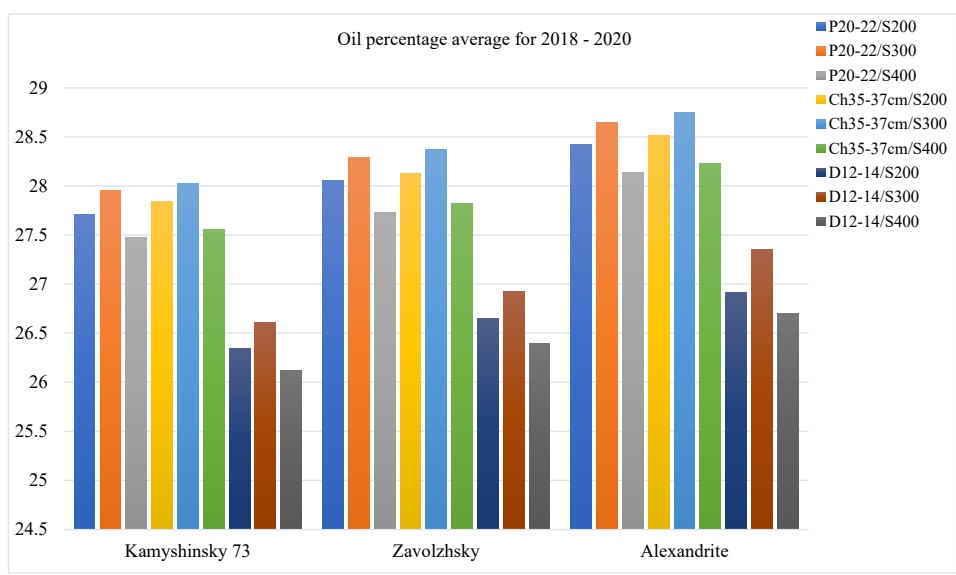

**Figure 5.** Interactive effect of different tillage techniques and seeding rates on average oil content. Abbreviations: P20–22/S200 represents moldboard plow PN-4-35 with depth of 20–22 cm/seeding rate of 200; P20–22/S300 represents moldboard plow PN-4-35 with depth of 20–22 cm/seeding rate of 300; P20–22/S400 represents moldboard plow PN-4-35 with depth of 20–22 cm/seeding rate of 400; Ch35–37 cm/S200 represents chisel plow OCHO 5-40 with depth of 35–37 cm/seeding rate of 200; Ch35–37 cm/S300 represents chisel plow OCHO 5-40 with depth of 35–37 cm/seeding rate of 300; Ch35–37 cm/S400 represents chisel plow OCHO 5-40 with depth of 35–37 cm/seeding rate of 400; D12–14/S200 represents disc plow BDM-4 with depth of 12–14 cm/seeding rate of 200; D12–14/S300 represents disc plow BDM-4 with depth of 12–14 cm/seeding rate of 300; and D12–14/S400 represents disc plow BDM-4 with depth of 12–14 cm/seeding rate of 400.

## 4. Discussion

In most cases, the best safflower yield and yield parameters, including dry biomass, leaf area and photosynthetic potential, were achieved when deep chisel tillage coupled with a seeding rate of 400 thousand seeds ha$^{-1}$ was used. The productivity of safflower depended on the varieties, tillage practices and plant density. In some regions of Russia, according to the soil conditions, the various tillage treatments have indicated a wide range of results with respect to the hydraulic conductivity of the agricultural soil. An analysis of the safflower yield and yield components showed that, on average, in the results obtained in 2018 to 2020, considering the tillage treatment, the highest yield was attained when chisel ploughing at a depth of 35–37 cm was performed. Our results were in congruence with findings from preceding studies [27,28] regarding the safflower growth and yield under different tillage systems. Wang et al. [27] reported that shallow tillage confers stable microbial colonies and nutrient utilization, thus enhancing soil properties and improving safflower growth and productivity. On the other hand, Paul et al. [29] recorded a higher yield with maximum soil disturbance and minimum yield with zero tillage. Wasaya et al. [30] and Steber et al. [31] evaluated the efficacy of various tillage systems on soil characteristics in their experiment and discovered that different tilling systems did not change the soil bulk density or hydraulic conductivity of loam soils.

Enhancing the seeding rate is one of the key tools of exploiting the safflower yield potential. Numerous researchers illustrated that different crop seeding rates significantly affected the safflower yield and yield component [32]. It can be inferred that the seeding rate can change the canopy structure and affect not just the intermodal elongation but also the thickening and accumulation of cell wall components in the safflower plant. In the majority of the cases observed in the study we carried out, the optimum results of the safflower yield components were attained by enhancing the safflower seeding rate. Emami et al. [33] reported that the oil content, seed yield and oil accumulation of safflower

significantly surged by increasing the seeding rate of safflower. Tarighi et al. [34] also stated that the lower safflower seeding rate resulted in a shorter safflower height in Iran. On the other hand, Elfadl et al. [16] did not observe the seeding rate efficacy in safflower oil accumulation in Germany. Yau [35] promulgated that the safflower plant height diminished by increasing the safflower seeding rate. Furthermore, there are numerous reports where the sowing rate increased when the seed yield was enhanced [36]. Bellé et al. [37] reported that the stem diameter of safflower plants significantly decreased during the winter and summer through the enhancement of the seeding rate to 128 seeds per $m^2$. Emami et al. [33] showed that the safflower seeding rate did not affect the 1000 seed weight during both seasons in their study. Amoughin et al. [38] and Shahri et al. [39] revealed that the oil accumulation also desirably improved with an increasing safflower seeding rate. On the other hand, Sharifi et al. [40] found that the oil percentage decreased by enhancing the seeding rate. In similar studies, Amoughin et al. [38] and Chakradhari et al. [41] also reported that increasing the safflower seeding rate significantly enhanced the yield and seed oil accumulation of safflower per hectare.

### 5. Conclusions

The safflower productivity depends on the variety, main tillage system and plant density. The maximum leaf area (26.35 $m^2/m^2$) and the highest photosynthetic potential (1489 thousand $m^2 \times day\ ha^{-1}$) were achieved when the Alexandrite cultivar was produced using deep chisel tillage and a seeding rate of 400 thousand seeds $ha^{-1}$. The highest safflower dry biomass was also attained in the Alexandrite cultivar using deep chisel tillage coupled with a 400 thousand seeds $ha^{-1}$ seeding rate. The best safflower yield (1.8 t $ha^{-1}$) and oil accumulation (28.7%) were achieved in the Alexandrite cultivar when deep chisel tillage and a sowing rate of 300 and 400 thousand seeds $ha^{-1}$ were used.

**Author Contributions:** Manuscript conception, S.V. and Y.P.; Methodology, M.Z.; Data analysis, M.Z.; Validation and investigation, S.K.; Writing—original draft preparation, M.Z. and S.K.; Writing—review and editing, M.Z.; Project administration, D.M. All authors have read and agreed to the published version of the manuscript.

**Funding:** This research received no external funding.

**Data Availability Statement:** The datasets used and/or analyzed during the current study are available from the corresponding author on reasonable request.

**Acknowledgments:** This work was supported by the RUDN University Strategic Academic Leadership Program.

**Conflicts of Interest:** The authors declare no conflict of interest.

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
