# Peer review of "Tillage System and Seeding Rate Impact on Yield, Oil Accumulation and Photosynthetic Potential of Different Cultivars of Safflower (Carthamus tinctorius L.) in Southern Russia"

_agronomy, doi:10.3390/agronomy12112904_

Round 1
Reviewer 1 Report
The manuscript is interesting as authors evaluated for three years the effect of tillage systems and seeding densities on some growth and yield characteristics of three safflower cultivars. The dataset is consistent. However, the description of what and how the experiments and evaluations were performed (material and methods section) and the presentation of the results (statistical analysis, tables and figures) are not adequate from my point of view. These issues need to be improved in the manuscript so that it can be accepted for publication. More comments in pdf file.

Author Response
Dear Reviewer
We gratefully acknowledge the detailed revision of the text and useful suggestions to improve the paper by the reviewer. We have closely followed he/she suggestions and introduced the required changes in the text. Main changes are addressed into the manuscript. Below, we have included reviewer comments and our responses.
We have revised the manuscript for edits/changes as you suggested. All the comments addressed in the text in YELLOW.
- Leaf area index m2/m2 addressed in entire text.
- We did not write effects of treatments on all cultivars as we are limited in 200 - 250 word, but we mentioned distinguished results on cultivars.
- Three repeated trial for each cultivar were performed.
- Double ridge stage signifies that the plant’s main growing point will produce no more primordial leaves, and instead will produce primordial spikelets of the young spike.
- The first sentence in Experimental design removed.
- Field preparation detail added to the M&M section. M&M part revised according to the reviewer comments. Data recording part also improved.
- Effect of experimental treatments on different cultivars were examined. Results in table 1 & 2; figure 1 & 5.
- Our table show average data for three years. In the capture of tables explained.
- Mean separation added to the tables.
- Each year, experiment (for each cultivar separate trial) with two treatments including basic tillage and seeding rates performed separately on three different safflower cultivar at the same area and the same field. The area under plots of each cultivar was 1,944 m2, and the total area under plots of whole three cultivars was 5,832 m2. As it is presented in line 115 – 117.
- We are not able to change type of figures and editing data because unfortunately our co-author who was responsible of this part has deceased in 2020 influenced by covid 19.
We hope that after these enhancements the manuscript can now be accepted for publication, although we are certainly willing to consider further changes if necessary.
Yours sincerely,
Reviewer 2 Report
Dear Authors,
congratulations on the draft article. The theme is good, but there are some things I suggest to change:
The form of the links does not meet MDPI requirements. Please correct this according to the form given on the website.
The literary references described in the draft article are good and appropriate to the topic, but they hardly include any of the newer literature that has been published in the last 5 years. Please add to this, because there have been many excellent works on the subject. It would be worthwhile to explain a little more the practical importance of Carthamus in agriculture, and to compare it in a few sentences with other field plants with a similar role.
The goals are not formulated or indicated.
In the Material and method chapter, no mention is made of the varieties used. Please do this. On the other hand, one is mentioned in the Abstract, but it is very confusing for the reader. In terms of easier interpretation of the Abstract, I recommend rewording it.
In addition, the Abstract can consist of a maximum of 200 words. Yours is currently 280 words. Please shorten it.
The terms cultivar and yield are too general for Keywords. Instead of these or next to them, it is worth adding more special words.
The data in the figures cannot be read accurately, the form of the diagram is not appropriate, and the name of the coordinates of every diagram is not legible. I recommend changing it.
In the Conclusions chapter, the results are briefly listed again. I recommend rewriting this chapter so that it is really the conclusions of the research.
The revision of the article is definitely necessary.
Author Response
Dear Reviewer
We gratefully acknowledge the detailed revision of the text and useful suggestions to improve the paper by the reviewer. We have closely followed he/she suggestions and introduced the required changes in the text. Main changes are addressed into the manuscript. Below, we have included reviewer comments and our responses.
We have revised the manuscript for edits/changes as you suggested. All the comments addressed in the text in YELLOW.
- Extensive English revision done through Track Changes Function.
- References and citation format in the text of the manuscript revised according to MDPI requirements.
- Several new citations added according to the reviewer suggestion, highlighted YELLOW in the references list.
- Objectives of the study revised at the end of Introduction part.
- We presented in the text of M&M in line 115 – 117 about safflower varieties in the study. Each year, experiment (for each cultivar separate trial) with two treatments including basic tillage and seeding rates performed separately on three different safflower cultivar at the same area and the same field. The area under plots of each cultivar was 1,944 m2, and the total area under plots of whole three cultivars was 5,832 m2.
- We are not able to change type of figures and editing data because unfortunately our co-author who was responsible of this part has deceased in 2020 influenced by covid 19. But we are ready to do if any special comment and suggestion you may have.
- Cultivars of safflower added to the abstract, and abstract revised in different aspects.
- Abstract reduced. If there is a need to shorten the abstract further, will definitely do it in the next steps.
- Keywords revised.
- Conclusion of the manuscript revised.
We hope that after these enhancements the manuscript can now be accepted for publication, although we are certainly willing to consider further changes if necessary.
Yours sincerely,
Round 2
Reviewer 2 Report
Dear Authors,
I acccept the manuscript. Congratulations on your work!
Author Response
Dear reviewer
Thanks for accepting our final version of manuscript